# Charge-Order on the Triangular Lattice: A Mean-Field Study for the Lattice *S* = 1/2 Fermionic Gas

**DOI:** 10.3390/nano11051181

**Published:** 2021-04-30

**Authors:** Konrad Jerzy Kapcia

**Affiliations:** Faculty of Physics, Adam Mickiewicz University in Poznań, ulica Uniwersytetu Poznańskiego 2, PL-61614 Poznań, Poland; konrad.kapcia@amu.edu.pl; Tel.: +48-61-829-5051

**Keywords:** charge order, triangular lattice, extended Hubbard model, atomic limit, mean-field theory, phase diagram, longer-range interactions, thermodynamic properties, fermionic lattice gas, adsorption on the surface

## Abstract

The adsorbed atoms exhibit tendency to occupy a triangular lattice formed by periodic potential of the underlying crystal surface. Such a lattice is formed by, e.g., a single layer of graphane or the graphite surfaces as well as (111) surface of face-cubic center crystals. In the present work, an extension of the lattice gas model to S=1/2 fermionic particles on the two-dimensional triangular (hexagonal) lattice is analyzed. In such a model, each lattice site can be occupied not by only one particle, but by two particles, which interact with each other by onsite *U* and intersite $W_{1}$ and $W_{2}$ (nearest and next-nearest-neighbor, respectively) density-density interaction. The investigated hamiltonian has a form of the extended Hubbard model in the atomic limit (i.e., the zero-bandwidth limit). In the analysis of the phase diagrams and thermodynamic properties of this model with repulsive $W_{1}>0$, the variational approach is used, which treats the onsite interaction term exactly and the intersite interactions within the mean-field approximation. The ground state (T=0) diagram for $W_{2}\leq 0$ as well as finite temperature (T>0) phase diagrams for $W_{2}=0$ are presented. Two different types of charge order within $\sqrt{3}\times \sqrt{3}$ unit cell can occur. At T=0, for $W_{2}=0$ phase separated states are degenerated with homogeneous phases (but T>0 removes this degeneration), whereas attractive $W_{2}<0$ stabilizes phase separation at incommensurate fillings. For $U/W_{1}<0$ and $U/W_{1}>1/2$ only the phase with two different concentrations occurs (together with two different phase separated states occurring), whereas for small repulsive $0<U/W_{1}<1/2$ the other ordered phase also appears (with tree different concentrations in sublattices). The qualitative differences with the model considered on hypercubic lattices are also discussed.

## 1. Introduction

It is a well known fact that the classical lattice gas model is useful phenomenological model for various phenomena. It has been studied in the context of experimental studies of adsobed gas layers on crystaline substrates (cf., for example pioneering works [1,2,3,4]). For instance, the adsorbed atoms exhibit tendency to occupy a triangular lattice formed by periodic potential of the underlying crystal surface. This lattice is shown in Figure 1a. Such a lattice is formed by, e.g., a single layer of graphane or the graphite surface [i.e., the honeycomb lattice; (0001) hexagonal closed-packed (hcp) surface], and (111) face-centered cubic (fcc) surface. Atoms from (111) fcc surface are organized in the triangular lattice, whereas the triangular lattice is a dual lattice for the honeycomb lattice [5]. Note also that arrangements of atoms on (110) base-centered cubic (bcc) surface as well as on (111) bcc surface (if one neglects the interactions associated with other layers under surface) are quite close to the triangular lattice. One should mention that the triangular lattice and the honeycomb lattice are two examples of two-dimensional hexagonal Bravais lattices. Formally, the triangular lattice is a hexagonal lattice with a one-site basis, whereas the honeycomb lattice is a hexagonal lattice with a two-site basis. The classical lattice gas model is equivalent with the S=1/2 Ising model in the external field [1,6,7,8,9] (the results for this model on the triangular lattice will be discussed in more details in Section 2).

In the present work, an extension of the lattice gas model to S=1/2 fermionic particles is analyzed. Such a model has a form of the atomic limit of the extended Hubbard model [10], cf. (Equation 1). In this model, each lattice site can be occupied not by only one particle as in the model discussed in previous paragraph, but also by two particles. In addition to long-range (i.e., intersite) interactions between fermions, the particles located at the same site can also interact with each other via onsite Hubbard *U* interaction. For a description of the interacting fermionic particles on the lattice, the single-orbital extended Hubbard model with intersite density-density interactions has been used widely [11,12,13,14,15,16,17,18,19]. It is one of the simplest model capturing the interplay between the Mott localization (onsite interactions) and the charge-order phenomenon [15,16,17,18,19,20,21,22,23,24,25]. However, in some systems the inclusion of other interactions and orbitals is necessary [11,12,13,14,26,27].

This work can be palced among recent theoretical and experimental studies of adsorption of various atoms on (i) the (0001) hcp surface of the graphite [28,29,30,31,32] or of other materials [33,34] and (ii) the (111) fcc surface of metals and semimetals [34,35,36,37,38,39]. Although in the mentioned works the adsorbed particles on surface are rather classical and the analysis of classical lattice gas can give some predictions, taking into account of the quantum properties of adsorbed particles is necessary for, e.g., a description of experiments with He${}^{4}$ and He${}^{3}$ [28,40,41]. Moreover, there is plethora of recent experimental and theoretical studies of quasi-two-dimensional systems, e.g., Na${}_{x}$CoO${}_{2}$ [42], NbSe${}_{2}$ [43,44,45,46,47], TiSe${}_{2}$ [48], TaSe${}_{2}$ [49], VSe${}_{2}$ [50], TaS${}_{2}$ [51], and other transition metals dichalcogenides [52] as well as organic conductors [53,54], where various charge-ordered patterns have been observed on the triangular lattice. However, for such phenomena the atomic limit of the model studies is less reliable and more realistic description includes also electron hoping term as in the extended Hubbard model [15,16,17,18,19] or coupling with phonons as in the Holstein-Hubbard model [55]. In such cases, results obtained for atomic limit can be treated as a benchmark for models including the itinerant properties of fermionic particles.

The present work is organized as follows. In Section 2 the model and the methods (together with the most important equations) are presented. Section 3 is devoted to the discussion of ground state phase diagrams of the model with non-zero next-nearest neighbor interactions. Next, the finite temperature properties of the model with only the nearest-neighbor interactions are presented in Section 4. Finally, the most important conclusions and supplementary discussion are included in Section 5.

## 2. The Model and the Method

The extended Hubbard model in the zero-bandwidth limit (i.e., in the atomic limit) with interactions restricted to the second neighbors (or, equivalently, the next-nearest neighbors) can be expressed as: (1)$$\begin{array}{ccc} \hat{H} & = & U\sum_{i}{\hat{n}}_{i↑}{\hat{n}}_{i↓}+\frac{1}{2}\frac{W_{1}}{z_{1}}\sum_{{\langle i,j\rangle }_{1}}{\hat{n}}_{i}{\hat{n}}_{j}+\frac{1}{2}\frac{W_{2}}{z_{2}}\sum_{{\langle i,j\rangle }_{2}}{\hat{n}}_{i}{\hat{n}}_{j}-\mu \sum_{i}{\hat{n}}_{i}, \end{array}$$
where ${\hat{n}}_{i}=\sum_{\sigma }{\hat{n}}_{i\sigma }$, ${\hat{n}}_{i\sigma }={\hat{c}}_{i\sigma }^{†}{\hat{c}}_{i\sigma }$, and ${\hat{c}}_{i\sigma }^{†}$ (${\hat{c}}_{i\sigma }$) denotes the creation (annihilation) operator of an electron with spin σ at the site *i*. *U* is the onsite density interaction, $W_{1}$ and $W_{2}$ are the intersite density-density interactions between the nearest neighbors (NN) and the next-nearest neighbors (NNNs), respectively. $z_{1}$ and $z_{2}$ are numbers of NN and NNNs, respectively. μ is the chemical potential determining the total concentration *n* of electrons in the system by the relation $n=\left(1/L\right)\sum_{i}\langle {\hat{n}}_{i}\rangle$, where 0≤n≤2 and *L* is the total number of lattice sites. In this work phase diagrams emerging from this model are inspected. The analyses are performed in the grand canonical ensemble.

In this work the mean-field decoupling of the intersite term is used in the following form
(2)$${\hat{n}}_{i}{\hat{n}}_{j}=\langle {\hat{n}}_{i}\rangle {\hat{n}}_{j}+{\hat{n}}_{i}\langle {\hat{n}}_{j}\rangle -\langle {\hat{n}}_{i}\rangle \langle {\hat{n}}_{j}\rangle ,$$
which is an exact treatment only in the limit of large coordination number ($z_{n}\to \infty$; or limit of infinite dimensions) [10,11,12,13,56,57,58]. Thus, for the two-dimensional triangular lattice (with $z_{1}=z_{2}=6$) it is an approximation in the general case. It should be underlined that the treatment of the onsite term is rigorous in the present work. Please note that that the interactions *U*, $W_{1}$ and $W_{2}$ should be treated as effective parameters for fermionic particles including all possible contributions and renormalizations originating from other (sub-)systems.

Model (Equation 1) for $W_{2}\neq 0$ has been intensively studied on the hypercubic lattices (see, e.g., [59,60,61,62,63,64,65,66,67,68] and references therein). Also the case of two-dimensional square lattice was investigated in detail for $W_{2}=0$ [61,62,63,64] as well as for $W_{2}\neq 0$ [65,66,67,68,69]. There are also rigorous results for one-dimensional chain for $W_{2}=0$ [70,71] and $W_{2}\neq 0$ [72].

In [73] the model with $W_{2}=0$ was investigated on the triangular lattice at half-filling by using a classical Monte Carlo method, and a critical phase, characterized by algebraic decay of the charge correlation function, belonging to the universality class of the two-dimensional XY model with a $Z_{6}$ anisotropy was found in the intermediate-temperature regime. Some preliminary results for model (Equation 1) on the triangular lattice and for large attractive U<0 and $W_{2}=0$ within the mean-field approximation were presented in [74].

The model in the limit U→−∞ is equivalent with the S=1/2 Ising model with antiferromagnetic (ferromagnetic) $J_{n}$ interactions if $W_{n}$ interaction in model (Equation 1) are repulsive, i.e., $W_{n}>0$ (attractive, i.e., $W_{n}<0$, respectively). The relation between interaction parameters in both models is very simple, namely $J_{n}=-W_{n}$. There is plethora of the results obtained for the Ising model on the triangular lattice. One should mention the following works (not assuming a comprehensive review): (a) exact solution in the absence of the external field *H*, i.e., for H=0 (only with NN interactions, at arbitrary temperature) [5,75,76,77,78,79]; (b) for the model with NNN interactions included: ground state exact results [2], Bethe-Peierls approximation [1], Monte Carlo simulation both for H=0 [80] and H≠0 [3] (and other methods, e.g., [81,82]); (c) exact ground state results for the model with up to 3rd nearest-neighbor interactions for both H=0 case [83] and H≠0 case [4,84]. The most important information arising from these analyses is that only for $W_{2}\leq 0$ (and arbitrary $W_{1}$) one can expect that consideration of $\sqrt{3}\times \sqrt{3}$ unit cells (i.e., tri-subblatice orderings) is enough to find all ordered states (particle arrangements) in the model. The reason is that the range of $W_{2}$ interaction is larger than the size of the unit cell. Thus, this is the point for that the present analysis of the model including only $\sqrt{3}\times \sqrt{3}$ unit cell orderings with restriction to $W_{2}\leq 0$ is justified. One should not expect occurrence of any other phases beyond the tri-sublattice assumption in the studied range of the model parameters.

Please note that for $W_{2}>0$ it is necessary to consider a larger unit cell to find the true phase diagram of the model even in the U→−∞ limit (cf., e.g., [2,4,81]). This is a similar situation as for model (Equation 1) on the square lattice, where for $W_{2}>0$ and any *U* not only checker-board order occurs (the two-sublattice assumption), but also other different arrangements of particles are present (the four-sublattice assumption, e.g., various stripes orders) [67,68].

### 2.1. General Definitions of Phases Existing in the Investigated System

In the systems analyzed only three nonequivalent homogeneous phases can exist (within the tree-sublattice assumption used). They are determined by the relations between concentrations $n_{\alpha }$’s in each sublattice α ($n_{\alpha }=\left(3/L\right)\sum_{i\in \alpha }\langle {\hat{n}}_{i}\rangle$), but a few equivalent solutions exist due to change of sublattice indexes. For intuitive understanding of rather complicated phase diagrams each pattern is marked with adequate abbreviation. The nonordered (NO) phase is defined by $n_{A}=n_{B}=n_{C}$ (all three $n_{\alpha }$’s are equal), the charge-ordered phase with two different concentrations in sublattices (DCO phase) is defined by $n_{A}=n_{B}\neq n_{C}$, $n_{B}=n_{C}\neq n_{A}$, or $n_{C}=n_{A}\neq n_{B}$ (two and only two out of three $n_{\alpha }$’s are equal, 3 equivalent solution), whereas in the charge-ordered phase with three different concentrations in sublattices (TCO phase) $n_{A}\neq n_{B}$, $n_{B}\neq n_{C}$, and $n_{A}\neq n_{C}$ (all three $n_{\alpha }$’s are different, 6 equivalent solutions). All these phases are schematically illustrated in Figure 1b. These phases exist in several equivalent solutions due to the equivalence of three sublattices forming the triangular lattice. Each of these patterns can be realized in a few distinct forms depending on specific electron concentrations on each sublattice (cf. Table 1 and Table 2 for T=0). In addition, the degeneracy of the ground state solutions is contained in Table 1 (including charge and spin degrees of freedom).

### 2.2. Expressions for the Ground State

In the ground state (i.e., for T=0), the grand canonical potential $\omega_{0}$ per site of model (Equation 1) can be found as
(3)$$\omega_{0}=\langle \hat{H}\rangle /L=E_{D}+E_{W}+E_{\mu },$$
where contributions associated with the onsite interaction, the intersite interactions, and the chemical potential, respectively, has the following forms
(4)$$\begin{array}{ccc} E_{D} & = & \frac{U}{6}\left[n_{A}\left(n_{A}-1\right)+n_{B}\left(n_{B}-1\right)+n_{C}\left(n_{C}-1\right)\right],\; \end{array}$$
(5)$$\begin{array}{ccc} E_{W} & = & \frac{1}{6}W_{1}\left(n_{A}n_{B}+n_{B}n_{C}+n_{C}n_{A}\right)+\frac{1}{6}W_{2}\left(n_{A}^{2}+n_{B}^{2}+n_{C}^{2}\right), \end{array}$$
(6)$$\begin{array}{ccc} E_{\mu } & = & -\frac{1}{3}\mu \left(n_{A}+n_{B}+n_{C}\right). \end{array}$$

In the above expressions, concentrations $n_{\alpha }$ at T=0 take the values from {0,1,2} set (cf. also Table 1). Please note that the above equations are the exact expressions for $\omega_{0}$ of model (Equation 1) on the triangular lattice.

The free energy per site of homogeneous phases at T=0 within the mean-field approximation is obtained as
(7)$$f_{0}=\langle \hat{H}+\mu \sum_{i}{\hat{n}}_{i}\rangle /L=UD_{occ}+E_{W},$$
where $E_{W}$ is expressed by (5). $D_{occ}=\left(1/L\right)\sum_{i}\langle {\hat{n}}_{i↑}{\hat{n}}_{i↓}\rangle$ denotes the double occupancy and this quantity is found to be exact, cf. Table 2. One should underline that above expression for $f_{0}$ is an approximate result for model (Equation 1) on the triangular lattice. Here, it is assumed that concentration $n_{\alpha }$ are as defined in Table 2 and they are the same in each $\sqrt{3}\times \sqrt{3}$ unit cell in the system. Formally, it could be treated as exact one only if the numbers $z_{n}$ (n=1,2) goes to infinity.

The expressions presented in this subsection (for $W_{2}=0$) can be obtained as the T→0 limit of the equations for T>0 included in Section 2.3.

### 2.3. Expressions for Finite Temperatures

For finite temperatures (T>0), the expressions given in [59] for the three-sublattice assumption takes the following forms (cf. also these in [68] given for the four-sublattice assumption). In approach used, the onsite *U* term is treated exactly and for the intersite $W_{1}$ term the mean-field approximation (Equation 2) is used. For a grand canonical potential ω (per lattice site) in the case of the lattice presented in Figure 1 one obtains
(8)$$\omega =-\frac{1}{6}\sum_{\alpha }\Phi_{\alpha }n_{\alpha }-\frac{1}{3\beta }\sum_{\alpha }\left(\ln Z_{\alpha }\right).$$
where $\beta =1/(k_{B}T)$ is inverted temperature, coefficients $\Phi_{\alpha }$ are defined as $\Phi_{\alpha }=\mu -\mu_{\alpha }$,
(9)$$Z_{\alpha }=1+2\exp\left(\beta \mu_{\alpha }\right)+\exp\left[\beta \left(2\mu_{\alpha }-U\right)\right],$$
and $\mu_{\alpha }$ is a local chemical potential in α sublattice (α∈{A,B,C})
(10)$$\begin{array}{c} \mu_{A}=\mu -\frac{1}{2}W_{1}\left(n_{B}+n_{C}\right),\;\mu_{B}=\mu -\frac{1}{2}W_{1}\left(n_{A}+n_{C}\right),\;\mu_{C}=\mu -\frac{1}{2}W_{1}\left(n_{A}+n_{B}\right). \end{array}$$

For electron concentration $n_{\alpha }$ in each sublattice in arbitrary temperature T>0 one gets
(11)$$n_{\alpha }=\frac{2}{Z_{\alpha }}\left\{\exp\left(\beta \mu_{\alpha }\right)+\exp\left[\beta \left(2\mu_{\alpha }-U\right)\right]\right\}\;\left(\mathrm{for}\;\alpha \in \left\{A,B,C\right\}\right).$$

The set of three Equations (Equation 11) for $n_{A}$, $n_{B}$, and $n_{C}$ determines the (homogeneous) phase occurring in the system for fixed model parameters *U*, $W_{1}$, and μ. If $n=\left(1/3\right)\left(n_{A}+n_{B}+n_{C}\right)$ is fixed, one has also set of three equations, but it is solved with respect to μ, $n_{A}$, and $n_{B}$ (the third $n_{\alpha }$ is obviously found as $n_{C}=3n-n_{A}-n_{B}$).

The free energy *f* per site is derived as
(12)$$f=\omega +\frac{1}{3}\mu \left(n_{A}+n_{B}+n_{C}\right),$$
where ω and $n_{\alpha }$’s are expressed by (Equation 8)–(Equation 11).

### 2.4. Macroscopic Phase Separation

The free energy $f_{PS}$ of the (macroscopic) phase separated state (as a function of total electron concentration *n*; and at any temperature T≥0) is calculated from
(13)$$f_{PS}\left(n\right)=\frac{n-n_{-}}{n_{+}-n_{-}}f_{+}\left(n_{+}\right)+\frac{n_{+}-n}{n_{+}-n_{-}}f_{-}\left(n_{-}\right),$$
where $f_{\pm }\left(n_{\pm }\right)$ are free energies of separating homogeneous phases with concentrations $n_{\pm }$. The factor before $f_{\pm }\left(n_{\pm }\right)$ is associated with a fraction of the system, which is occupied by the phase with concentration $n_{\pm }$. Such defined phase separated states can exist only for *n* fulfilling the condition $n_{-}<n<n_{+}$. For $n_{\pm }$ only the homogeneous phase exists in the system (one homogeneous phase occupies the whole system). Concentrations $n_{\pm }$ are simply determined at the ground state, whereas for T>0 they can be found as concentrations at the first-order (discontinuous) boundary for fixed μ or by minimizing the free energy $f_{PS}$ [i.e., (Equation 13)] with respect to $n_{+}$ and $n_{-}$ (for *n* fixed). For more details of the so-called Maxwell’s construction and macroscopic phase separations see, e.g., [59,68,85,86]. The interface energy between two separating phases is neglected here.

## 3. Results for the Ground State ($W_{1}>0$ and $W_{2}\leq 0$)

### 3.1. Analysis for Fixed Chemical Potential μ

The ground state diagram for model (Equation 1) as a function of (shifted) chemical potential $\bar{\mu }=\mu -W_{1}-W_{2}$ is shown in Figure 2. The diagram is determined by comparison of the grand canonical potentials $\omega_{0}$’s of all phases collected in Table 1 [cf. (Equation 3)]. It consists of several regions, where the NO phase occurs (3 regions: NO${}_{0}$, NO${}_{1}$ and NO${}_{2}$), the DCO phase occurs (6 regions: DCO${}_{1}$, DCO${}_{2}$, DCO${}_{3}$, DCO${}_{1}^{*}$, DCO${}_{2}^{*}$, and DCO${}_{3}^{*}$) and the TCO phase occurs (1 region).

All boundaries between the phases in Figure 2 are associated with a discontinuous change of at least one of the $n_{\alpha }$. The only boundaries associated with a discontinuous jump of two $n_{\alpha }$’s are: DCO${}_{2}$–DCO${}_{3}$ (DCO${}_{2}^{*}$–DCO${}_{3}^{*}$) and TCO–NO${}_{1}$. At the boundaries $\omega_{0}$’s of the phases are the same. It means that both phases can coexist in the system provided that a formation of the interface between two phases does not require additional energy. For $W_{2}=0$, only the boundaries DCO${}_{2}$–DCO${}_{3}$ (DCO${}_{2}^{*}$–DCO${}_{3}^{*}$) and TCO–NO${}_{1}$ have finite degeneracy (6 and 7, respectively, modulo spin degrees of freedom) and the interface between different types of $\sqrt{3}\times \sqrt{3}$ unit cells increases the energy of the system. Thus, the mentioned phases from neighboring regions cannot coexist at the boundaries. The other boundaries exhibit infinite degeneracy (it is larger than $3\cdot 2^{L/3}$ modulo spin) and entropy per site in the thermodynamic limit is non-zero. It means that at these boundaries both types of unit cells from neighboring regions can mix with any ratio and the formation of the interface between two phases does not change energy of the system. However, some conditions for arrangement of the cells can exist. For example, the DCO${}_{2}$ phase with (0,0,2) can mix with the DCO${}_{2}^{*}$ phase with (0,2,2) or (2,0,2), but not with the DCO${}_{2}^{*}$ phase with (2,2,0). Please note that it is also possible to mix all three unit cells: (0,0,2), (0,2,2), and (2,0,2). In such a case, (0,2,2) and (2,0,2) cells of the DCO${}_{2}^{*}$ phase cannot be located next to each other, i.e., they need to be separated by (0,0,2) unit cells of the DCO${}_{2}$ phase. Thus, the degeneracy of the DCO${}_{2}$–DCO${}_{2}^{*}$ boundary is indeed larger than $3\cdot 2^{L/3}$ modulo spin. This is so-called *macroscopic degeneracy*, cf. [68]). In such a case, we say that the *microscopic phase separation* occurs. For $W_{2}<0$ these degeneracies are removed and all boundaries exhibit finite degeneracy (neglecting spin degrees of freedom). In this case the phases cannot be mixed on a microscopic level.

Please also note that for $W_{2}=0$ as well as for $W_{2}<0$ inside the regions shown in Figure 2, the $\sqrt{3}\times \sqrt{3}$ unit cells of the same type with different orientation cannot mix. It denotes that orientation of one type of the unit cell determines the orientation of other unit cells (of the same type). Thus, the degeneracy of the state of the system is finite (modulo spin) and the system exhibits the long-range order at the ground state inside each region of Figure 2. This is different from the case of two dimensional square lattice, where inside some regions different unit cells (elementary blocks) of the same phase can mix with each other [67,68].

One should underline that the discussed above ground state results for fixed chemical potential are the exact results for model (Equation 1) on the triangular lattice. This is due to the fact that the model is equivalent with a classical spin model, namely the S=1 Blume-Cappel model with two-fold degenerated value of S=0 (or the S=1 classical Blume-Cappel with temperature-dependent anizotropy without degeneration), cf. [10,60,63]. For such a model, the mean-field approximation is an exact theory at the ground state and fixed external magnetic field (which corresponds to the fixed chemical potential in the model investigated).

### 3.2. Analysis for Fixed Particle Concentration *n*

The ground state diagram as a function of particle concentration *n* is shown in Figure 3. The rectangular regions are labeled by the abbreviations of homogeneous phases (cf. Table 2). At commensurate filling, i.e., i/3 (i=0,1,2,3,4,5,6; but only on the vertical boundaries indicated in Figure 3) the homogeneous phase occurs, which can be found in Table 1 and Figure 2. On the horizontal boundaries the phases from both neighboring regions have the same energies.

For $W_{2}=0$ phase separated states (mentioned in the last column of Table 2) are degenerated with the corresponding homogeneous phases inside all regions of the phase diagram. This degeneracy can be removed in finite temperatures and in some regions the phase separated states can be stable at T>0 (such regions are indicated by slantwise patter in Figure 3, cf. also Section 4). E.g., for $W_{2}=0$, the TCO phase can exist only in the range of $0<U/W_{1}<1/2$ at T≠0. For $W_{2}<0$ the phase separated states have lower energies and they occur on the phase diagram (inside the rectangular regions of Figure 3). Obviously, at commensurate filling and for any $W_{2}\leq 0$, the homogeneous states can only occur (i.e., solid vertical lines in Figure 3). Please note that the following boundaries between homogeneous states (obtained by comparing only energies of homogeneous phases): (i) the DCO${}_{A}$ and DCO${}_{B}$ phases, (ii) the DCO${}_{A}$ and DCO${}_{C}$ phases, and (iii) the TCO${}_{A}$ and TCO${}_{B}$ phases are located at $U/W_{1}=0$ (and these corresponding for n>1; the dashed line in Figure 3). For $W_{2}<0$ these lines do not overlap with the boundaries between corresponding phase separated states at $U/W_{1}-\left|k\right|=0$ (or $U/|W_{2}|=1$), but in such a case the homogeneous states have higher energies than the phase separated states. In fact, the homogeneous states for $W_{2}<0$ are unstable (i.e., ∂μ/∂n<0) inside the regions of Figure 1. For $W_{2}<0$ they are stable only for commensurate fillings (solid lines in Figure 3).

For the system on the square lattice the similar observation can be made (Figure 1 from [59])—compare HCO${}_{A}$–LCO${}_{A}$ and HCO${}_{A}$–HCO${}_{B}$ boundaries at $U/W_{1}=0$ with PS1${}_{A}$–PS1${}_{B}$ and PS1${}_{A}$–PS1${}_{B}$ boundaries at $U/|W_{2}|=1$, respectively. In [68] the boundaries between homogeneous phases for $W_{2}<0$ are not shown in Figure 3. Only boundaries between corresponding phase separated states are correctly presented in that figure for $W_{2}<0$. For $U/W_{1}>0$, the CBO${}_{A}$ phase (corresponding to the HCO${}_{A}$ phase from [59]) is not the phase with the lowest energy (among homogeneous phases) in any range of *n* (but for $U/W_{1}<0$ it has the lowest energy among all homogeneous states). However, the corresponding phase separated state NO${}_{0}$/CBO${}_{2}$ (i.e., PS1${}_{A}$ from [59]) can occur for $U/W_{1}>0$ (and for $U/|W_{2}|<1$) as shown in Figure 3 of [68].

The vertical boundaries for homogeneous phases (i.e., the transitions with changing *n*) are associated with continuous changes of all $n_{\alpha }$’s and $D_{occ}$, but the chemical potential μ (calculated as μ=∂f/∂n) changes discontinuously. Boundaries DCO${}_{A}$–DCO${}_{B}$, DCO${}_{A}$–DCO${}_{C}$, and TCO${}_{A}$–TCO${}_{B}$ (and other transitions for fixed *n* at $U/W_{1}=0$) between homogeneous phases are associated with discontinuous change of only $D_{occ}$. One should note that it is similar to transition between two checker-board ordered phases on the square lattice, namely CBO${}_{A}$–CBO${}_{B}$ and CBO${}_{A}$–CBO${}_{C}$ boundaries, cf. [68] (or the HCO${}_{A}$–LCO${}_{A}$ and HCO${}_{A}$–HCO${}_{B}$ boundaries, respectively, from [59]). At the other horizontal boundaries (i.e., transitions for fixed *n* at $U/W_{1}-\left|k\right|=1/2$ in Figure 3) two of $n_{\alpha }$’s and $D_{occ}$ change discontinuously. At commensurate fillings transitions with changing $U/W_{1}$ occur only at points indicated by squares in Figure 3.

All horizontal boundaries between phase separated states (which are stable for $W_{2}<0$) are connected with discontinuous changes of $D_{occ}$. These boundaries located at $U/W_{1}-\left|k\right|=0$ are also associated to a discontinuous change of particle concentration in one of the domains.

The diagram presented in Figure 3 is constructed by the comparison of (free) energies of various homogeneous phases and phase separated states collected in Table 1. The energies of homogeneous phases are calculated from (Equation 7), whereas energies of phase separated states are calculated from (Equation 13). Please note that it is easy to calculate energies of $f_{\pm }\left(n_{\pm }\right)$ of separating homogeneous phase (with commensurate fillings) at the ground state by just taking μ=0 in $\omega_{0}$’s collected in Table 1. Obviously, one can also calculate energies of the phases collected in Table 2 at these fillings (from both neighboring regions). For example, the DCO${}_{B}$ phase and the DCO${}_{C}$ phase at n=1/3 reduce to DCO${}_{1}$ phase.

## 4. Results for Finite Temperatures ($W_{1}>0$ and $W_{2}=0$)

One can distinguish four ranges of *U* interaction, where the system exhibits qualitatively different behavior, namely: (i) $U/W_{1}<0$, (ii) $0<U/W_{1}<\left(1/3\right)\ln\left(2\right)$, (iii) $\left(1/3\right)\ln\left(2\right)<U/W_{1}<1/2$, and (iv) $U/W_{1}>1/2$. In Figure 4, Figure 5, Figure 6 and Figure 7, the exemplary finite temperature phase diagrams occurring in each of these ranges of onsite interaction are presented. All diagrams are found by investigation of the behavior of $n_{\alpha }$’s determined by (Equation 11) in the solution corresponding to the lowest grand canonical potential [Equation (Equation 8), when μ is fixed] or to the lowest free energy [Equations (Equation 12) and (Equation 13) if *n* is fixed]. The set of three nonlinear Equations (Equation 11) has usually several nonequivalent solutions and thus it is extremely important to find a solution, which has the minimal adequate thermodynamic potential. In Figure 8 the behavior of $n_{\alpha }$’s as a function of temperature or chemical potential is shown for some representative model parameters. Figure 9 presents the phase diagram of the system for half-filling.

For $U/W_{1}<0$ and $U/W_{1}>1/2$ the phase diagrams of the model are similar and the DCO phase is only ordered homogeneous one occurring on the diagrams. In the first range, there are two regions of ordered phase occurrence (cf. Figure 4 and [74]), whereas in the second case one can distinguish four regions of the DCO phase stability (cf. Figure 5). The NO–DCO transitions for fixed μ are discontinuous for any values of onsite interaction and chemical potential in discussed range of model parameters and thus phase separated state PS${}_{1}$:NO/DCO occurs in define ranges of *n*. In this state domains of the NO and the DCO phases coexist.

For $U/W_{1}<0$ the temperature of NO–DCO transition is maximal for $\bar{\mu }=0$ (i.e., at half-filling)–Figure 4a. Its maximal value $T_{M}$ monotonously decreases with increasing of *U* from $k_{B}T_{M}/W_{1}=1/2$ for U→−∞ and at U=0 it is equal to 1/4. This transition exhibits re-entrant behavior (for fixed $|\bar{\mu }|>1$). At $T=T_{M}$ and $\bar{\mu }=0$ and at only this point, this transition exhibits properties of a second order transition (cf. Figure 8a). In particular, with increasing *T* for fixed $\bar{\mu }=0$$n_{\alpha }$’s changes continuously at $T_{M}$, but two equivalent solutions still exist for any $T<T_{M}$ (similarly as in the ferromagnetic Ising model at zero field [9]). At $\bar{\mu }=0$ and $T<T_{M}$ the discontinuous transition between two DCO phases occurs. In the DCO phase for $\bar{\mu }<0$ (n<1) [connecting with the DCO${}_{1}$ (DCO${}_{A}$) region at T=0] the relation $n_{A}=n_{B}<n_{C}$ is fulfilled, whereas in the DCO phase for $\bar{\mu }>0$ (n>1) [connecting with the DCO${}_{1}^{*}$ (DCO${}_{A}^{*}$) region at T=0] the relation $n_{A}<n_{B}=n_{C}$ occurs ($n_{C}$ can be larger than 1 for some temperatures), cf. also Figure 8g,h as well as [74]. Both discontinuous transitions for fixed chemical potential are associated with occurrence of phase separated states. On the diagrams obtained for fixed *n* three region of phase separated states occurs (Figure 4b). For $W_{2}=0$ the PS${}_{1}$:NO/DCO phase separated state occurs only for T>0. For T→0 the concentrations in both domains of the PS${}_{1}$ state approach 0 (or 2), whereas for $T\to T_{M}$ they approach to 1. Near n=1 the PS${}_{2}$:DCO/DCO state is stable for $0\leq T<T_{M}$. In this state domains of two DCO phases (with different particle concentrations) coexist in the system.

For $U/W_{1}>1/2$ the diagrams are similar, but the double occupancy of sites is strongly reduced due to repulsive *U* (Figure 5). Thus, their structure exhibits two lobs of the DCO phase occurrence in cotrary to the case of $U/W_{1}<0$, where a single lob of the DCO phase is present (as expected from previous studies of the model, cf. [10,59,60]). The maximal value $k_{B}T_{M}/W_{1}$ of NO–DCO transition occurs for $\bar{\mu }/W_{1}$ corresponding approximately quarter fillings (i.e., near n=1/2 and n=3/2). With increasing *U* it decreases and finally in the limit U→+∞ it reaches 1/8. At this point DCO–NO boundary exhibits features of continuous transition as discussed previously. In this range, the phase diagrams are (almost) symmetric with respect to these fillings (when one considers only one part of the diagram for 0<n<1 or for 1<n<2).

The most complex diagrams are obtained for $0<U/W_{1}<1/2$, where the TCO phase appears at T=0 and for finite temperatures near half-filling. For $0<U/W_{1}<\left(1/3\right)\ln\left(2\right)$ the region of the TCO phase is separated from the NO phase by the region of DCO phase, Figure 6a. The TCO–DCO transition is continuous (cf. Figure 8g,h for $U/W_{1}=0.35$) and its maximal temperature is located for half-filling (at $\bar{\mu }=0$ or n=1). At this point two first-order NO–DCO and two second-order TCO–DCO boundaries merge (for fixed chemical potential). It is the only point for fixed $U/W_{1}$ in this range of model parameters, where a direct continuous transition from the TCO phase to the NO phase is possible (Figure 8b). The continuous TCO–DCO transition temperature can be also found as a solution of (Equation 11) and (Equation 17) as discussed in Appendix A. Similarly as for $U/W_{1}<0$, the temperature of NO–DCO transition is maximal at half-filling. For fixed *n*, the narrow regions of PS${}_{1}$:NO/DCO states are present between the NO region and DCO regions. Please note that for T>0 there is no signatures of the discontinuous DCO${}_{1}$–DCO${}_{2}$ (DCO${}_{1}^{*}$–DCO${}_{2}^{*}$) boundary occurring at T=0. It is due to the fact that the discontinuous jumps of $n_{\alpha }$’s occurring for T=0 at these boundaries are changed into continuous evolutions of sublattice concentrations at T>0 and there is no criteria for distinction of these two DCO phases at finite temperatures (cf. also [59,60,61]). From the same reason, there is no boundary at T>0 for fixed *n* associated to the DCO${}_{B}$–DCO${}_{C}$ (DCO${}_{B}^{*}$–DCO${}_{C}^{*}$) line occurring at T=0 (Figure 6b). However, strong reduction of one $n_{\alpha }$ from the case where $n_{\alpha }\approx 2$ to the case of $n_{\alpha }\approx 1$ is visible (some kind of a smooth crossover inside the DCO region), cf. Figure 8f–h for $U/W_{1}=0.35$.

For $\left(1/3\right)\ln\left(2\right)<U/W_{1}<1/2$, the maximum of the NO–DCO transition temperature is shifted towards larger $|\bar{\mu }|/W_{1}$ (or smaller |1−n|). This is associated with forming of the two-lob structure of the diagram found for $U/W_{1}>1/2$. Inside the regions of the DCO phase occurrence discontinuous transitions between two DCO phases appear—See Figure 7a as well as Figure 8e,i. These new regions of the DCO phase at T>0 [with $n_{A}<n_{B}=n_{C}$ (for $\bar{\mu }<0$ or n<1); cf. Figure 8e,i] are connected with the DCO${}_{3}$ and DCO${}_{3}^{*}$ regions occurring at the ground state. The boundaries DCO–DCO weakly dependent on $\bar{\mu }$ are associated with occurrence of phase separated PS${}_{2}$:DCO/DCO states (at high temperatures) in some ranges of *n*, cf. Figure 7b. The other DCO–DCO transitions (which are almost temperature-independent) are not connected with phase separated states. Also the first-order TCO–NO line is present near half-filling, cf. Figure 8d. One should underline that all four lines (three first-order boundaries: DCO–NO, DCO–DCO, TCO–NO and the second-order TCO–DCO boundary) merge at single point with numeric accuracy. However, it cannot be excluded that the DCO–NO and TCO–DCO boundaries connect with the temperature-independent line in slightly different points, what could result in, e.g., the TCO–DCO–NO sequence of transition with increasing temperature for small range of chemical potential $\bar{\mu }$. All of these almost temperature-independent boundaries (i.e., the DCO–DCO and the TCO–NO lines) are located at temperature, which decreases with increasing $U/W_{1}$ and approaches 0 at $U/W_{1}=1/2$ [i.e., they connect with the DCO${}_{2}$–DCO${}_{3}$ (DCO${}_{2}^{*}$–DCO${}_{3}^{*}$) and TCO–NO${}_{1}$ boundaries at T=0 for fixed μ or with the TCO–DCO${}_{D}$ (TCO${}^{*}$–DCO${}_{D}^{*}$) lines at T=0 for fixed *n*]. From the analysis of (Equation 11) similarly as it was done in the case of the square lattice [10] (see also Appendix A) one obtains that the point, where the TCO–NO transition changes its order at half-filling, is $k_{B}T/W_{1}=1/6$ and $U/W_{1}=\left(1/3\right)\ln\left(2\right)$.

For better overview of the system behavior, the phase diagram of the model for half-filling ($\bar{\mu }=0$ or n=1) is presented in Figure 9. The temperature of the order-disorder transition decreases with increasing $U/W_{1}$. In low temperatures and for $U/W_{1}<0$, the DCO phases exist in the system (precisely, if μ is fixed—at $\bar{\mu }=0$ the DCO–DCO discontinuous boundary occurs; whereas if *n* is fixed—the PS${}_{2}$:DCO/DCO state is stable at n=1), cf. also Figure 4. For $0<U/W_{1}<1/2$ the TCO phase is stable below the order-disorder line, but for $\left(1/3\right)\ln\left(2\right)<U/W_{1}<1/2$ and $k_{B}T/W_{1}<1/6$ the TCO–NO phase transition is discontinuous (cf. also Figure 7). For $U/W_{1}<\left(1/3\right)\ln\left(2\right)$ the order-disorder boundary presented in Figure 9 is a merging point of several boundaries as presented in Figure 4 and Figure 6, and discussed previously. Thus, formally this order-disorder boundary for $U/W_{1}<\left(1/3\right)\ln\left(2\right)$ occurring at half-filling is a line of some critical points of a higher order.

Please note that the order-disorder transition is discontinuous for any value of onsite interaction and chemical potential [excluding only the TCO–NO boundary for half-filling and $0<U/W_{1}<\left(1/3\right)\ln\left(2\right)$] in contrast to the case of two- [10,59,60] or four-sublattice [67,68] assumptions, where it can be continuous one for some range of model parameters). In [74] also metastable phases have been discussed in detail for the large onsite attraction limit and the triangular lattice.

## 5. Final Remarks

In this work, the mean-field approximation was used to investigate the atomic limit of extended Hubbard model [hamiltonian (Equation 1)] on the triangular lattice. The phase diagram was determined for the model with intersite repulsion between the nearest neighbors ($W_{1}>0$). The effects of attractive next-nearest-neighbor interaction ($W_{2}<0$) were discussed in the ground state. The most important findings of this work are that (i) two different arrangements of particles (i.e., two different charge-ordered phases: the DCO and TCO states) can occur in the system and (ii) attractive $W_{2}<0$ or finite T>0 removes the degeneration between homogeneous phases and phase separated states occurring at T=0 for $W_{2}=0$. It was shown that TCO phase is stable in intermediate range of onsite repulsion $0<U/W_{1}<1/2$ (for $W_{2}=0$). All transition from the ordered phases to the NO are discontinuous for fixed chemical potential (apart from TCO–NO boundary at half-filling for $0<U/W_{1}<\left(1/3\right)\ln\left(2\right)$) and the DCO–NO boundaries at single points corresponding to n=1/2,1,3/2 as discussed in Section 4), thus the phase separated states occur on the phase diagram for fixed particle concentration.

One should stress that hamiltonian (Equation 1) is interesting not only from statistical point of view as a relatively simple toy model for phase transition investigations. Although it is oversimplified for quantitative description of bulk condensed matter systems, it can be useful in qualitative analysis of, e.g., experimental studies of adsorbed gas layers on crystalline substrates.

Additionally, one notes that the mean-field results for model (Equation 1) with attractive $W_{1}<0$ and $W_{2}\leq 0$ are the same for both two-sublattice and tri-sublattice assumptions. In such a case, three different nonordered phases exist with the discontinuous first-order transition between them (at $\bar{\mu }=0$ for U<0 or for $|\bar{\mu }|\neq 0$ for $U/(|W_{1}|+|W_{2}|)>1$), and thus for fixed *n*, several so-called electron-droplet states (phase separations NO/NO) exist (cf. [60,68,87,88], particularly Figure 2 of [60]).

Notice that the mean-field decoupling of the intersite term is an approximation for purely two-dimensional model investigated, which overestimates the stability of ordered phases. For example, the order-disorder transition for the ferromagnetic Ising model is overestimated by the factor two (for the honeycomb, square and triangular lattices rigorous solution gives $k_{B}T_{c}/\left|J\right|$ as 0.506, 0.568, 0.607, respectively, whereas the mean-field approximation gives $k_{B}T_{c}/\left|J\right|=1$) [76]. Moreover, the results for the antiferromagnetic Ising model on the triangular lattice [the limit U→±∞ of model (Equation 1)] do not predict long-range order at zero field [1,3,76] and T>0 [corresponding to n=1 or n=1/2,3/2, respectively, in the case of model (Equation 1)]. However, longer-range interactions [3] or weak interactions between adsorbed particles and the adsorbent material occurring in realistic systems could stabilize such an order (such systems are rather quasi-two-dimensional). It should be also mentioned that the charge Berezinskii-Kosterlitz-Thouless-like phase was found in the intermediate-temperature regime between the charge-ordered phase (with long-range order, coresponding to the TCO phase here) and disordered phases in the investigated model [73].

The recent progress in the field of optical lattices and a creation of the triangular lattice by laser trapping [89,90] could enable testing predictions of the present work. The fermionic gases in harmonic traps are fully controllable systems. Note also that the superconductivity in the twisted-bilayer graphene [91,92,93,94,95,96] is driven by the angle between the graphene layers. It is associated with an occurrence of the Moiré pattern (the triangular lattice with very large supercell). Hetero-bilayer transition metals dichalcogenides system is the other field where this pattern appears [97,98]. This makes further studies of properties of different models on the triangular lattice desirable.

## Figures and Tables

**Figure 1 nanomaterials-11-01181-f001:**
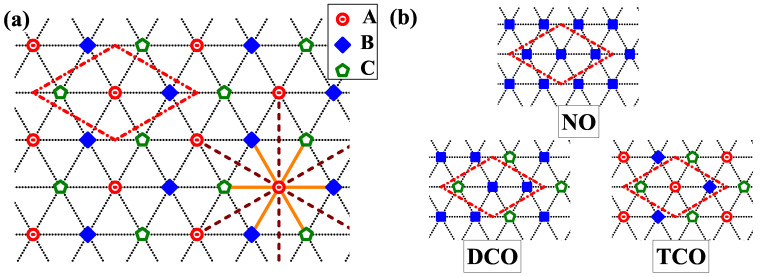
(**a**) The schema of the triangular lattice on which the extended Hubbard model in the atomic limit is studied in the present work. The lattice is divided into three equivalent sublattices (α=A,B,C) denoted by different symbols. The dash-dotted line denotes the boundaries of $\sqrt{3}\times \sqrt{3}$ unit cell. By solid and dashed lines all nearest neighbors and all next-nearest neighbors of a chosen site from sublattice *A* are indicated, respectively. (**b**) There different types of particle arrangements in $\sqrt{3}\times \sqrt{3}$ unit cells (i.e., the tri-sublattice assumption) corresponding to NO, DCO, TCO phases (as labeled). Symbol shapes on each panel correspond to respective concentrations at the lattice sites.

**Figure 2 nanomaterials-11-01181-f002:**
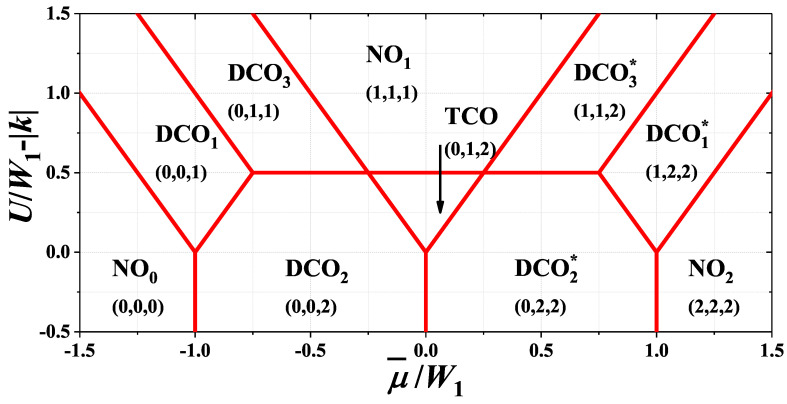
Ground state phase diagram of the model on the triangular lattice as a function of shifted chemical potential $\bar{\mu }=\mu -W_{1}-W_{2}$ for $W_{1}>0$ and $W_{2}\leq 0$ ($\left|k|=\right|W_{2}|/W_{1}$). The regions are labeled by the names of the phases defined in Table 1 and numbers corresponding to concentrations in each sublattice $n_{A}$, $n_{B}$ and $n_{C}$.

**Figure 3 nanomaterials-11-01181-f003:**
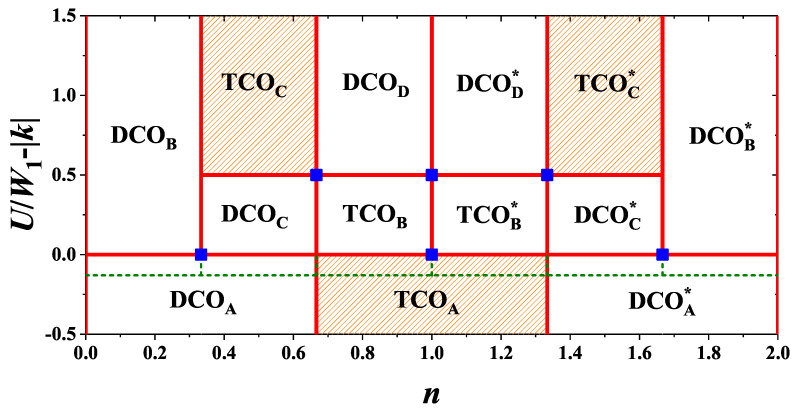
Ground state phase diagram of the model as a function of particle concentration *n* for $W_{1}>0$ and $W_{2}\leq 0$ ($\left|k|=\right|W_{2}|/W_{1}$). The regions are labeled by the names of the homogeneous phases (cf. Table 2). For $W_{2}=0$ all homogeneous phases are degenerated with macroscopic phase separated states indicated in the last column of Table 2. In regions filled by slantwise pattern the phase separated states occurs at infinitesimally T>0 for $W_{2}=0$. For $W_{2}<0$ the phase separated states occur inside the regions, whereas at the vertical boundaries for commensurate filling the homogeneous states (defined in Table 1) still exist. The boundary at $U/W_{1}=0$ (schematically indicated by dashed green line) denotes the boundaries between homogeneous phases, which do not overlap with the boundaries between phase separated states for $W_{2}<0$. Squares denote transitions for fixed *n* between homogeneous phase at commensurate fillings.

**Figure 4 nanomaterials-11-01181-f004:**
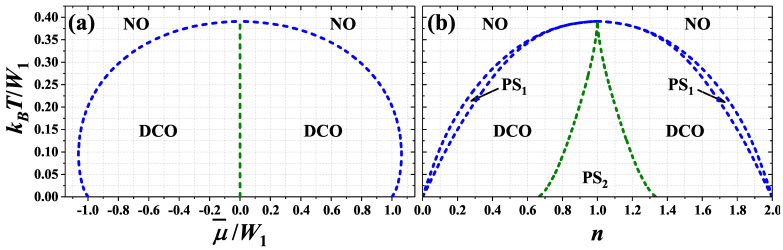
Phase diagrams of the model for $U/W_{1}=-1.00$ as a function of (**a**) chemical potential $\bar{\mu }/W_{1}$ and (**b**) particle concentration *n* ($W_{1}>0$, $W_{2}=0$). All transitions are first order and regions of phase separated state (PS${}_{1}$:NO/DCO and PS${}_{2}$:DCO/DCO) occurrence are present on panel (**b**). NO and DCO denote homogeneous phases defined in Figure 1b.

**Figure 5 nanomaterials-11-01181-f005:**
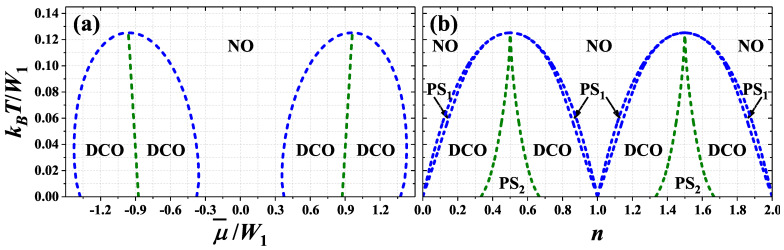
Phase diagrams of the model for $U/W_{1}=0.75$ as a function of (**a**) chemical potential $\bar{\mu }/W_{1}$ and (**b**) particle concentration *n* ($W_{1}>0$, $W_{2}=0$). All transitions are first order. Other denotations as in Figure 4.

**Figure 6 nanomaterials-11-01181-f006:**
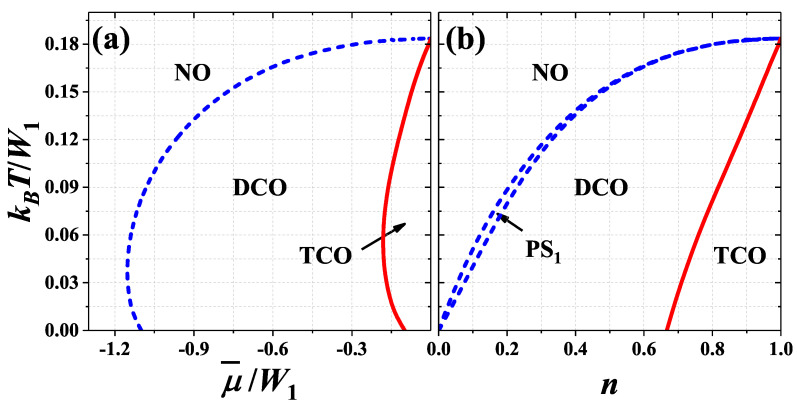
Phase diagrams of the model for $U/W_{1}=0.20$ as a function of (**a**) chemical potential $\bar{\mu }/W_{1}$ and (**b**) particle concentration *n* ($W_{1}>0$, $W_{2}=0$). The boundary TCO–DCO is second order, the remaining are first order. Other denotations as in Figure 4. The diagrams are shown only for $\bar{\mu }\leq 0$ and n≤1, but they are symmetric with respect to $\bar{\mu }=0$ and n=1, respectively.

**Figure 7 nanomaterials-11-01181-f007:**
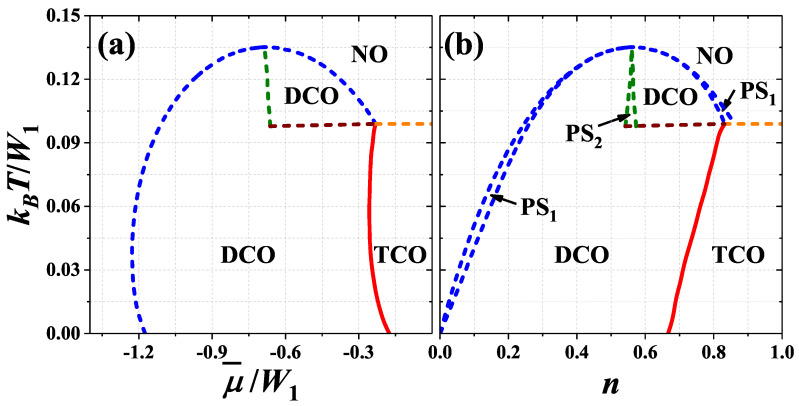
Phase diagrams of the model for $U/W_{1}=0.35$ as a function of (**a**) chemical potential $\bar{\mu }/W_{1}$ and (**b**) particle concentration *n* ($W_{1}>0$, $W_{2}=0$). The boundary TCO–DCO is second order, the remaining are first order. Other denotations as in Figure 4. The diagrams are shown only for $\bar{\mu }\leq 0$ and n≤1, but they are symmetric with respect to $\bar{\mu }=0$ and n=1, respectively.

**Figure 8 nanomaterials-11-01181-f008:**
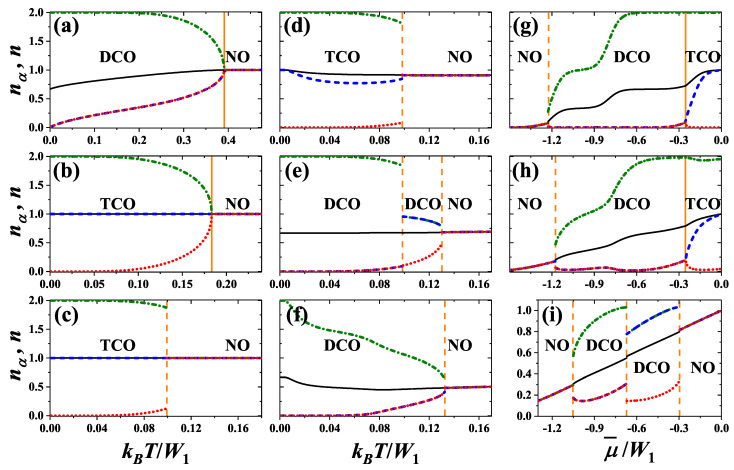
Dependencies of particle concentrations $n_{\alpha }$’s in the sublattices (red dotted, blue dashed, and green dot-dashed lines) as a function of $k_{B}T/W_{1}$ [(**a**–**f**)] and $\bar{\mu }/W_{1}$ [(**g**–**h**)] for $W_{2}=0$. Black solid lines denote total particle concentration $n=(n_{A}+n_{B}+n_{C})/3$. They are obtained for: (**a**) $U/W_{1}=-1.00$, $\bar{\mu }/W_{1}=0$; (**b**) $U/W_{1}=0.20$, $\bar{\mu }/W_{1}=0$; (**c**) $U/W_{1}=0.35$, $\bar{\mu }/W_{1}=0$; (**d**) $U/W_{1}=0.35$, $\bar{\mu }/W_{1}=-0.15$; (**e**) $U/W_{1}=0.35$, $\bar{\mu }/W_{1}=-0.5$; (**f**) $U/W_{1}=0.35$, $\bar{\mu }/W_{1}=-0.8$; (**g**) $U/W_{1}=0.35$, $k_{B}T/W_{1}=0.40$; (**h**) $U/W_{1}=0.35$, $k_{B}T/W_{1}=0.80$; (**i**) $U/W_{1}=0.35$, $k_{B}T/W_{1}=0.11$. Vertical solid and dashed lines indicate points of continuous and discontinuous transitions, respectively.

**Figure 9 nanomaterials-11-01181-f009:**
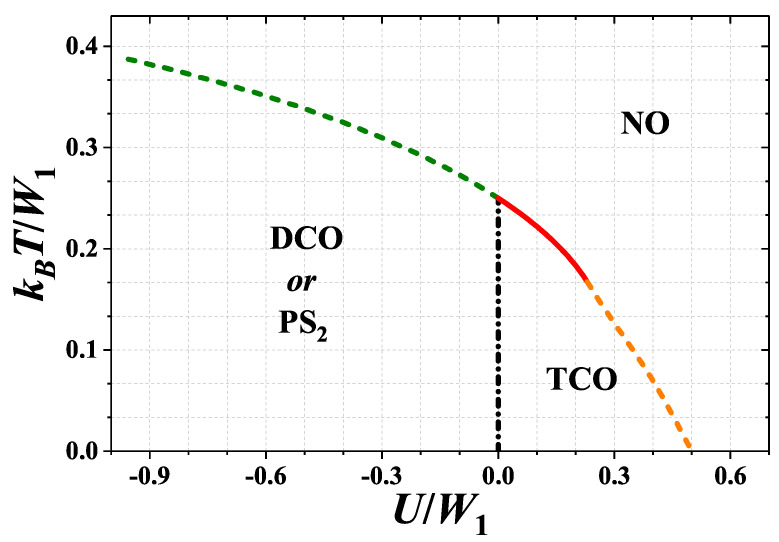
Phase diagram of the model for half-filling ($\bar{\mu }=0$ or n=1) as a function of onsite interaction $U/W_{1}$ ($W_{1}>0$ and $W_{2}=0$). The order-disorder boundary for $U/W_{1}<\left(1/3\right)\ln\left(2\right)$ is a line consisting of some higher-order critical points as discussed in the text.

**Table 1 nanomaterials-11-01181-t001:** Homogeneous phases ($z_{n}\to \infty$, n=1,2) or $\sqrt{3}\times \sqrt{3}$ unit cells (the triangular lattice) at T=0 (for fixed μ). Star “*” in superscript indicates that the phase is obtained by the particle-hole transformation (i.e., $n_{\alpha }\to 2-n_{\alpha }$; the NO${}_{1}$ and TCO phases are invariant under this transformation). In the brackets also an alternative name is given. The degeneration $d_{c}\times d_{s}$ of the unit cells (equal to the degeneration of the ground state for $z_{n}\to \infty$ limit) and degeneration $D_{c}\times D_{s}$ of the ground state phases constructed from the corresponding unit cells for the triangular lattice is given (with respect to charge and spin degrees of freedom).

Phase	$n_{A}$	$n_{B}$	$n_{C}$	$d_{c}\times d_{s}$	$D_{c}\times D_{s}$	$\omega_{0}$
NO${}_{0}$ (NO${}_{2}^{*}$)	0	0	0	1×1	1×1	0
NO${}_{1}$ (NO${}_{1}^{*}$)	1	1	1	1×8	$1\times 2^{L}$	$(-2\mu +W_{1}+W_{2})/2$
NO${}_{2}$ (NO${}_{0}^{*}$)	2	2	2	1×1	1×1	$-2\mu +U+2W_{1}+2W_{2}$
DCO${}_{1}$	0	0	1	3×2	$3\times 2^{L/3}$	$(-2\mu +W_{2})/6$
DCO${}_{1}^{*}$	1	2	2	3×2	$3\times 2^{L/3}$	$(-10\mu +4U+8W_{1}+9W_{2})/6$
DCO${}_{2}$	0	0	2	3×1	3×1	$(-2\mu +U+2W_{2})/3$
DCO${}_{2}^{*}$	0	2	2	3×1	3×1	$(-4\mu +2U+2W_{1}+4W_{2})/3$
DCO${}_{3}$	0	1	1	3×4	$3\times 4^{L/3}$	$(-4\mu +W_{1}+2W_{2})/6$
DCO${}_{3}^{*}$	1	1	2	3×4	$3\times 4^{L/3}$	$(-8\mu +2U+5W_{1}+6W_{2})/6$
TCO (TCO${}^{*}$)	0	1	2	6×2	$6\times 2^{L/3}$	$(-6\mu +2U+2W_{1}+5W_{2})/6$

**Table 2 nanomaterials-11-01181-t002:** Homogeneous phases at T=0 (for fixed *n*) defined by $n_{\alpha }$’s and $D_{occ}$. $n_{s}$ and $n_{f}$ define the range $\left[n_{s},n_{f}\right]$ of *n*, where the phase is correctly defined. In the last column, the phase separated state degenerated with the homogeneous phase in range $\left(n_{s},n_{f}\right)$ for $W_{2}=0$ is mentioned. Star “*” in superscript indicates that the phase is obtained by the particle-hole transformation (i.e., $n_{\alpha }\to 2-n_{\alpha }$; TCO${}_{A}$, TCO${}_{A}^{*}$, TCO${}_{B}$, and TCO${}_{B}^{*}$ phases are invariant under this transformation).

Phase	$n_{A}$	$n_{B}$	$n_{C}$	$D_{\mathrm{occ}}$	$n_{s}$	$n_{f}$	PS
DCO${}_{A}$	0	0	3n	n/2	0	2/3	NO${}_{0}$/DCO${}_{2}$
DCO${}_{B}$	0	0	3n	0	0	1/3	NO${}_{0}$/DCO${}_{1}$
DCO${}_{C}$	0	0	3n	n−1/3	1/3	2/3	DCO${}_{1}$/DCO${}_{2}$
DCO${}_{D}$	3n−2	1	1	0	2/3	1	DCO${}_{3}$/NO${}_{1}$
TCO${}_{A}$	0	3n−2	2	n/2	2/3	4/3	DCO${}_{2}$/DCO${}_{2}^{*}$
TCO${}_{B}$	0	3n−2	2	1/3	2/3	1	DCO${}_{2}$/TCO
TCO${}_{C}$	0	3n−1	1	0	1/3	2/3	DCO${}_{1}$/DCO${}_{3}$
DCO${}_{A}^{*}$	3n−4	2	2	n/2	4/3	2	DCO${}_{2}^{*}$/NO${}_{2}$
DCO${}_{B}^{*}$	3n−4	2	2	n−1	5/3	2	DCO${}_{1}^{*}$/NO${}_{2}$
DCO${}_{C}^{*}$	3n−4	2	2	2/3	4/3	5/3	DCO${}_{2}^{*}$/DCO${}_{1}^{*}$
DCO${}_{D}^{*}$	1	1	3n−2	n−1	1	4/3	NO${}_{1}$/DCO${}_{3}^{*}$
TCO${}_{A}^{*}$	0	3n−2	2	n/2	2/3	4/3	DCO${}_{2}$/DCO${}_{2}^{*}$
TCO${}_{B}^{*}$	0	3n−2	2	n−2/3	1	4/3	TCO${}^{*}$/DCO${}_{2}^{*}$
TCO${}_{C}^{*}$	1	3n−3	2	n−1	4/3	5/3	DCO${}_{3}^{*}$/DCO${}_{1}^{*}$

## Data Availability

The data presented in this study are available on request from the author. All data presented in Section 4 have been obtained by numerical solving of self-consistent equations given in Section 2.3.

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
