# Peer review of "Charge-Order on the Triangular Lattice: A Mean-Field Study for the Lattice S = 1/2 Fermionic Gas"

_nanomaterials, 2021, doi:10.3390/nano11051181_

Round 1

Reviewer 1 Report

Referee report

Title: Charge-Order on the Triangular Lattice: a Mean-Field Study for the Lattice S = 1/2 Fermionic Gas

By K.J. Kapcia

This is an interesting theoretical manuscript on the charge-order on a triangular lattice. The author has used a mean-field approximation to scrutinize the extended Hubbard model on a triangular lattice. On each lattice site two particles can be placed, which interact with each other via an on-site interaction (U). In addition, the author also included nearest (W1) and next-nearest neighbor interactions (W2) between the particles. The author has determined the phase diagram of this triangular lattice using the aforementioned extended Hubbard model. As far as I can judge the presented results are solid and scientifically sound. Unfortunately, the manuscript contains quite a number of textual errors and numerous sentences do not read well.

As an example I take the first sentence of the abstract:

The adsorbed atoms exhibit tendency to lie on a triangular lattice formed by periodic potential of the underlying crystal surface such as a single layer of graphane or the graphite  surfaces as well as (111) surface of face-cubic center crystals.

I propose to change this sentence to:

On lattices with a triangular periodicity, such as graphene and the (111) surfaces of  face-cubic center crystals, adsorbed atoms prefer to adopt this triangular structure.

Throughout the manuscript there are many order of these sentences that do not read well and therefore I strongly encourage the authors to proofread their manuscript and/or consults a person with a good command of the English language.

In summary, this is an interesting manuscript. Before this manuscript can be accepted for publication a thorough revision of the text is required.

Author Response

Dear Reviewer 1,

I uploaded this reply file as well as a new version of my manuscript with all the corresponding changes I have made. All modification are marked by the red text in pdf file provided for the review purposes.

Comment: This is an interesting theoretical manuscript on the charge-order on a triangular lattice. The author has used a mean-field approximation to scrutinize the extended Hubbard model on a triangular lattice. On each lattice site two particles can be placed, which interact with each other via an on-site interaction (U). In addition, the author also included nearest (W1) and next-nearest neighbor interactions (W2) between the particles. The author has determined the phase diagram of this triangular lattice using the aforementioned extended Hubbard model. As far as I can judge the presented results are solid and scientifically sound. Unfortunately, the manuscript contains quite a number of textual errors and numerous sentences do not read well.

Reply: I would like to thank the Referee for positive review and useful suggestions. The manuscript has been consulted with a person with a command of the English language and the text has been modified accordingly. I agree this is very important to present the results in the way which is easy understandable and clear for the potential reader.

Comment: As an example I take the first sentence of the abstract: “The adsorbed atoms exhibit tendency to lie on a triangular lattice formed by periodic potential of the underlying crystal surface such as a single layer of graphane or the graphite  surfaces as well as (111) surface of face-cubic center crystals. “ I propose to change this sentence to: “On lattices with a triangular periodicity, such as graphene and the (111) surfaces of  face-cubic center crystals, adsorbed atoms prefer to adopt this triangular structure.” Throughout the manuscript there are many order of these sentences that do not read well and therefore I strongly encourage the authors to proofread their manuscript and/or consults a person with a good command of the English language.

Reply: This sentence of the abstract has been corrected. The other changes have been introduced to improve the English language and to make the manuscript more easily understandable for a potential reader. All modifications are marked by red color in the pdf file for review purposes. I believe that other typos and language mistakes can be corrected after acceptance and during editing and proofreading process.

Comment: In summary, this is an interesting manuscript. Before this manuscript can be accepted for publication a thorough revision of the text is required.

Reply: I hope that all introduced changes in the manuscript improved the presentation of the results. I believe that the manuscript in the present version is ready for publication.

All modification are marked by the red text in pdf file provided for the review purposes. The major changes are as following:

1) A typo for the annihilation operator below Eq. 1 has been corrected (accordingly to the comment of Reviewer 2).

2) The definition of $n_\alpha$ has been added after equation (8) (accordingly to the comment of Reviewer 2).

3) The explanations of DCO and TCO abbreviations have been added in the third paragraph of Section 2 (accordingly to the comment of Reviewer 2). This paragraph has been moved before start of Section “Expressions for the ground state” and a new Section 2.1 has been created.

4) Figure 9 together with its discussion in one before last paragraph in Section 4 has been added for better overview of the results presented.

5) The text of the Appendix (two last paragraphs) has been modified to be more precise.

6) Reference [99] has been added.

7) Some other minor changes and language corrections have been introduced as highlighted by the color text (accordingly to suggestion of Referee 1).

Reviewer 2 Report

The author analyzes the distributions of fermions on hexagonal lattices in terms of the extended Hubbard model, Eq. 1, in the atomic (t = 0) limit. First and second neighbor interactions W1 and W2 are kept (Fig. 1) in addition to on-site U. Fermion distributions are obtained in the thermodynamic limit using the grand partition function, a mean-field approximation for W1, W2 at both constant chemical mu potential and constant fermion density n. As discussed in Section 2, the chosen model has unordered (UO), doubly charge ordered (DCO) and triply charge ordered (TCO) phases. Section 3 presents the ground-state phase boundaries for W1 > 0, W2 ≤ 0 at constant mu (Fig. 2) and constant n (Fig. 3), as well as the nature of the transitions. The rich T ≥ 0 phase diagrams in Section 4 for W1 > 1, W2 = 0 are shown in Figs. 5-8.

I recommend publication. This is a fine paper, carefully presented, with many references to related lattice gas models with only single occupancy at sites. Section 2 is particularly clear in stating the problems and the equations to be solved. Quite properly, only results are presented, and some features of the ground-state phase diagram can be readily understood from Hubbard models. I only focused on a few of the many cases in Section 4.

The acronyms DCO and TCO are defined at the end under “abbreviations”. They should be in the main text, where only UO appears. Also, typo for the annihilation operator below Eq. 1. Is the coefficient of n_alpha defined in Eq. 8? I missed it.

Author Response

Dear Reviewer 2,

I uploaded this reply as well as a new version of our manuscript with all the corresponding changes I have made. All modification are marked by the red text in pdf file provided for the review purposes.

At this point I would like to thank you for Your review with useful suggestions, and I hope that my reply and changes accounts to all Your issues.

Comment: The author analyzes the distributions of fermions on hexagonal lattices in terms of the extended Hubbard model, Eq. 1, in the atomic (t = 0) limit. First and second neighbor interactions W1 and W2 are kept (Fig. 1) in addition to on-site U. Fermion distributions are obtained in the thermodynamic limit using the grand partition function, a mean-field approximation for W1, W2 at both constant chemical mu potential and constant fermion density n. As discussed in Section 2, the chosen model has unordered (UO), doubly charge ordered (DCO) and triply charge ordered (TCO) phases. Section 3 presents the ground-state phase boundaries for W1 > 0, W2 ≤ 0 at constant mu (Fig. 2) and constant n (Fig. 3), as well as the nature of the transitions. The rich T ≥ 0 phase diagrams in Section 4 for W1 > 1, W2 = 0 are shown in Figs. 5-8.

Reply: I would like to thank the Referee for careful reading of the manuscript.

Comment: I recommend publication. This is a fine paper, carefully presented, with many references to related lattice gas models with only single occupancy at sites. Section 2 is particularly clear in stating the problems and the equations to be solved. Quite properly, only results are presented, and some features of the ground-state phase diagram can be readily understood from Hubbard models. I only focused on a few of the many cases in Section 4.

 Reply: I would like to thank the Referee for very positive review of the manuscript and recommendation for the publication. I also introduced some improvements in Section 4 for better readability for potential readers.

Comment: The acronyms DCO and TCO are defined at the end under “abbreviations”. They should be in the main text, where only UO appears.

Reply: Yes, I agree that they should appear in the main text. The explanations of these abbreviations  have been added in the third paragraph of Section 2. This paragraph has been moved before start of Section “Expressions for the ground state” and a new Section 2.1 has been created.

Comment: Also, typo for the annihilation operator below Eq. 1.

Reply: Indeed, there was a type. The dagger symbol has been removed.

Comment: Is the coefficient of n_alpha defined in Eq. 8? I missed it.

Reply: No, it was not defined. The definition of $n_\alpha$ has been added after equation (8). Also after equation (A4) in the appendix some definitions were missing and they have been added.

All modification are marked by the red text in pdf file provided for the review purposes. The major changes are as following:

1) A typo for the annihilation operator below Eq. 1 has been corrected (accordingly to the comment of Reviewer 2).

2) The definition of $n_\alpha$ has been added after equation (8) (accordingly to the comment of Reviewer 2).

3) The explanations of DCO and TCO abbreviations have been added in the third paragraph of Section 2 (accordingly to the comment of Reviewer 2). This paragraph has been moved before start of Section “Expressions for the ground state” and a new Section 2.1 has been created.

4) Figure 9 together with its discussion in one before last paragraph in Section 4 has been added for better overview of the results presented.

5) The text of the Appendix (two last paragraphs) has been modified to be more precise.

6) Reference [99] has been added.

7) Some other minor changes and language corrections have been introduced as highlighted by the color text (accordingly to suggestion of Referee 1).

Round 2

Reviewer 1 Report

The author has improved his manuscript. I recommend to accept this manuscript.